# D-Dimer/Fibrinogen Ratio and Radiological Severity Scores in Acute Pulmonary Embolism: Is There Room for a New Thrombus-Burden Marker?

**DOI:** 10.3390/diagnostics15222875

**Published:** 2025-11-13

**Authors:** Francesco Tiralongo, Lorenzo Musmeci, Stefania Tamburrini, Giacomo Sica, Mariano Scaglione, Mariapaola Tiralongo, Rosita Comune, Corrado Ini’, Davide Giuseppe Castiglione, Emanuele David, Pietro Valerio Foti, Stefano Palmucci, Antonio Basile

**Affiliations:** 1Radiology Unit 1, Department of Medical Surgical Sciences and Advanced Technologies “GF Ingrassia”, University Hospital Policlinico “G. Rodolico-San Marco”, University of Catania, 95123 Catania, Italy; lorenz.musmec@gmail.com (L.M.); corrado.ini@gmail.com (C.I.); davidegiuseppecastiglione@gmail.com (D.G.C.); david.emanuele@yahoo.it (E.D.); pietrofoti@hotmail.com (P.V.F.); basile.antonello73@gmail.com (A.B.); 2 Department of Radiology, Ospedale del Mare, ASL NA1 Centro, 80147 Naples, Italy; tamburrinistefania@gmail.com (S.T.); ros.comune@gmail.com (R.C.); 3Department of Radiology, Monaldi Hospital, Azienda Ospedaliera dei Colli, 80131 Naples, Italy; gsica@sirm.org; 4Department of Medicine, Surgery and Pharmacy, University of Sassari, 07100 Sassari, Italy; mscaglione@uniss.it; 5Department of Clinical and Experimental Medicine, University of Catania, 95123 Catania, Italy; m.tiralongo97@gmail.com; 6UOSD I.P.T.R.A. (Pulmonary and Advanced Radiological Techniques Unit), Department of Medical Surgical Sciences and Advanced Technologies “GF Ingrassia”, University Hospital Policlinico “G. Rodolico-San Marco”, University of Catania, 95123 Catania, Italy; spalmucci@unict.it

**Keywords:** pulmonary embolism, CT pulmonary angiography, right-ventricular dysfunction, D-dimer, fibrinogen, D-dimer/fibrinogen ratio

## Abstract

**Background/Objectives****:** The D-dimer/fibrinogen ratio (D-d/F) has been proposed as a composite marker of fibrinolysis–coagulation balance. Whether D-d/F reflects CT-quantified thrombus burden and right ventricular dysfunction (RVD) in acute pulmonary embolism (PE) remains uncertain. **Methods:** Single-center retrospective cohort of consecutive adults with CTPA-confirmed PE (January 2022–October 2024). D-d/F = D-dimer (µg/mL)/fibrinogen (mg/dL). Thrombus burden: Qanadli and Mastora indices. RVD: RV/LV ratio, septal bowing, and IVC reflux. Associations: Spearman’s ρ with Steiger’s Z for between-marker comparisons. Discrimination for Qanadli ≥ 40% and RV/LV ≥ 1.0 by ROC. Two exploratory logistic models predicted Qanadli ≥ 40%: Model-1 (age, sex, D-d/F) and Model-2 adding RV/LV. **Results:** Among 112 patients (mean age 65.4 ± 15.6; 60% men), D-d/F correlated modestly with Qanadli (ρ = 0.233, *p* = 0.013) and Mastora (ρ = 0.274, *p* = 0.0034); strengths were similar to D-dimer (no between-marker difference: Steiger’s Z both *p* > 0.5). D-d/F correlated with RV/LV (ρ = 0.335, *p* < 0.001) and with IVC reflux (ρ = 0.247, *p* = 0.0085). CT indices related more strongly to hemodynamic markers (e.g., Qanadli with RV/LV ρ = 0.571, *p* < 0.0001; Mastora with RV/LV ρ = 0.620, *p* < 0.0001). Patients with septal bowing had higher D-dimer (median 4.65 vs. 2.74 µg/mL, *p* = 0.0037), higher D-d/F (1.04 vs. 0.61, *p* = 0.0018), and higher clot-burden scores (both *p* < 0.0001). For Qanadli ≥ 40%, AUCs were 0.621 for D-d/F (cut-off > 0.795; sens 58.8%, spec 62.3%) and 0.618 for D-dimer (>1.894 µg/mL; 84.3%, 37.7%); AUCs did not differ (*p* = 0.93). For RV/LV ≥ 1.0, AUCs were 0.693 for D-d/F (>0.607; 83.8%, 52.0%) and 0.684 for D-dimer (>2.849 µg/mL; 75.7%, 54.7%); *p* = 0.72. In Model-1, D-d/F predicted Qanadli ≥ 40% (OR = 1.43 per unit, *p* = 0.043; AUC = 0.64). After adding RV/LV (Model-2), discrimination improved (AUC = 0.796), RV/LV remained a strong predictor (*p* < 0.0001), and D-d/F was not retained (*p* = 0.287). **Conclusions:** In acute PE, D-d/F tracks thrombus burden and RVD to a degree comparable to D-dimer, but effects are modest. CT-based markers—particularly RV/LV—better reflect disease severity and are more predictive of high clot burden. Risk prediction and incremental utility of D-d/F were not assessed and warrant prospective evaluation.

## 1. Introduction

Acute pulmonary embolism (PE) is a major cause of cardiovascular morbidity and mortality, with an estimated incidence of 39–115 cases per 100,000 person-years [1]. Clinical presentation is often nonspecific, making timely diagnosis and risk stratification essential to guide management and improve outcomes [1]. Computed tomography pulmonary angiography (CTPA) is currently the gold standard for PE diagnosis, allowing both direct visualization of endoluminal thrombi and quantitative assessment of clot burden and right ventricular (RV) strain [2,3].

Quantitative CT obstruction indices, such as the Qanadli and Mastora scores, provide an objective estimate of pulmonary arterial occlusion [4,5]. These indices have been shown to correlate with echocardiographic and hemodynamic parameters of RV dysfunction and are increasingly considered in prognostic stratification. RV strain, assessed by CT-derived parameters such as the RV-to-LV diameter ratio, interventricular septal bowing, and inferior vena cava (IVC) contrast reflux, is a robust predictor of adverse outcomes and is independently associated with short-term mortality in acute PE [2].

In parallel, laboratory biomarkers play an essential role in diagnosis and prognosis. D-dimer, a fibrin degradation product, is widely employed in PE work-up due to its high sensitivity. However, its low specificity—particularly in elderly patients or in the presence of malignancy, infection, or recent surgery—limits its utility as a standalone test [6,7,8]. Fibrinogen, an acute-phase reactant consumed during thrombus formation, may provide complementary information. The D-dimer/fibrinogen ratio (D-d/F) has been proposed as a biomarker integrating fibrinolytic activity and coagulation status.

D-d/F has a clear physiologic basis: D-dimer reflects fibrin breakdown, while fibrinogen—an acute-phase protein and clotting substrate—rises with inflammation and can fall with consumption during thrombosis [8]. Their ratio helps mute inflammatory noise on D-dimer and captures the balance between coagulation and fibrinolysis.

Preliminary studies, limited to small or selected cohorts (e.g., postpartum or ICU populations), suggest that this ratio may enhance the diagnostic accuracy of D-dimer and provide additional prognostic information in thromboembolic disease [8,9,10].

However, the association between D-d/F ratio, CT-quantified thrombotic load, and RV dysfunction in acute PE has not been comprehensively investigated. Establishing such a relationship could provide a simple laboratory surrogate to complement imaging, enabling earlier risk stratification in clinical practice.

The present study primarily aimed to evaluate whether the D-dimer/fibrinogen ratio varies according to thrombotic burden, as assessed by validated CT obstruction scores, and with CTPA signs of right ventricular strain in patients with acute pulmonary embolism, and to test whether these associations are stronger than those observed for D-dimer alone.

The secondary aim was to quantify the strength of the association between CT thrombus-burden indices and CTPA signs of RV dysfunction.

We hypothesized that the D-d/F ratio would correlate more strongly with CT thrombus burden and RVD than D-dimer alone.

## 2. Materials and Methods

This single-center observational study retrospectively evaluated patients with a diagnosis of acute pulmonary embolism (PE) confirmed by computed tomography pulmonary angiography (CTPA).

The study population consisted of consecutive adult patients with a diagnosis of acute pulmonary embolism (PE) confirmed by computed tomography pulmonary angiography (CTPA) at our institution between January 2022 and October 2024. Clinical, laboratory, and imaging data were retrieved from the institutional electronic medical records. Comorbidities, cardiac biomarkers (troponin/BNP), and clinical severity scores (PESI/sPESI) were not recorded by the registry and were not available for extraction.

The study was conducted in accordance with the Declaration of Helsinki. Ethics approval was not required for retrospective observational studies, without the use of medicinal products, based exclusively on fully anonymized, minimal-risk data, as set out in the Standard Operating Procedures of the Territorial Ethics Committee of the Sicilian Region (Version 4, 19 March 2025).

Inclusion criteria comprised the availability of a diagnostic CTPA performed within 24 h of emergency department admission and complete laboratory data, including D-dimer and fibrinogen levels, obtained at presentation. Patients were excluded if they had evidence of chronic or previously diagnosed pulmonary embolism, cerebrovascular disease, myocardial infarction, poor-quality CT scans precluding quantitative analysis, a history of advanced malignancy or end-stage disease or missing critical laboratory or imaging data. Patients under anticoagulant treatment before presentation were also excluded.

### 2.1. Population

A total of 112 patients met the inclusion criteria (Figure 1).

60 patients were males (53.57%) and 52 were females (46.42%); the mean age at presentation was 65.38 ± 15.59 years.

The characteristics of patients included in our analysis are summarized in Table 1.

### 2.2. Laboratory Parameters Evaluation

Laboratory parameters recorded at the time of diagnosis included D-dimer (ng/mL) and fibrinogen (mg/dL), both measured using the same automated analyzer in the institutional central laboratory.

At emergency department admission, venous blood (2 mL) was drawn within 6 h into 3.2% sodium citrate tubes (9:1). The samples were centrifuged at 3000 rpm for 5 min at room temperature to obtain citrated plasma. D-dimer was measured by a latex-enhanced immunoturbidimetric assay on an ACL TOP 300 analyzer (Instrumentation Laboratory/Werfen) and reported as fibrinogen-equivalent units (µg/mL FEU). Fibrinogen was determined using the Clauss method on the same platform and reported in milligrams per deciliter (mg/dL).

The D-dimer/fibrinogen ratio (D-d/F), calculated as (D-dimer (µg/mL)/(Fibrinogen (mg/dL)), was determined for each patient and served as the primary laboratory marker of interest.

### 2.3. CTPA Acquisition and Imaging Metrics

CTPA examinations were performed on a 128-slice multidetector CT scanner (GE Revolution, GE Healthcare, Milwaukee, WI, USA). All patients underwent a contrast-enhanced chest CT scan in the supine position with their arms raised. Scanning was performed in a caudo-cranial direction using a bolus-tracking technique. A region of interest (ROI) was placed in the main pulmonary artery, and acquisition was triggered automatically once the enhancement threshold of 80 Hounsfield units was reached. The contrast agent used was Iomeprol 400 (Iomeron^®^, Bracco Imaging, Milan, Italy), administered via a peripheral venous catheter at approximately 4 mL/s, resulting in an iodine delivery rate of 1.6 gI/s. The contrast volume ranged from 40 to 60 mL, depending on the flow rate, injection duration, and patient body weight, followed by a 40 mL saline flush.

All CT images were reconstructed with a slice thickness of 1 mm or less and analyzed on a dedicated workstation.

Pulmonary clot burden was quantified using both the Qanadli and Mastora scoring systems. The Qanadli score was calculated by dividing the pulmonary arterial tree into 10 segmental arteries per lung (3 in the upper lobe, 2 in the middle or lingular lobe, and 5 in the lower lobe). Partial and complete occlusions were scored as 1 and 2, respectively. For proximal emboli, the score was derived by summing the corresponding segmental branches distal to the occlusion. The maximum possible score was 40, and results were expressed as a percentage: Qanadli index = Σ (embolism number × degree of obstruction)/40 × 100% (Figure 2) [4].

The Mastora score was computed by evaluating five mediastinal, six lobar, and twenty segmental arteries, totaling 31 pulmonary branches. The degree of vascular obstruction was graded on a 5-point scale based on the percentage of luminal occlusion: 1 for <25%, 2 for 25–49%, 3 for 50–74%, 4 for 75–99%, and 5 for complete (100%) obstruction. The maximum possible score was 155, and the percentage index was calculated as: Mastora index = Σ (embolism grade × number of affected vessels)/155 × 100% (Figure 3) [5].

Right ventricular dysfunction (RVD) was evaluated based on three criteria: a right-to-left ventricular short-axis diameter ratio (RV/LV) measured on the same axial slice at the valvular plane, using inner-edge-to-inner-edge calipers, so that both cavities were sampled in the same cardiac phase (normal value < 1), interventricular septal bowing toward the left ventricle (deviation of the interventricular septum was evaluated as follows: normal (convex toward the RV), flattened, and septal bowing (convex toward the left ventricle), and contrast reflux into the inferior vena cava (IVC) graded ≥ 2 on a four-point scale. The main pulmonary artery diameter (PA) and ascending aorta (Ao) diameter were also recorded, and the PA/Ao ratio was calculated (Figure 4).

All CTPA scans were independently reviewed by two radiologists with 5 and 10 years of experience, respectively. All image readers were blinded to all clinical and laboratory results, including D-d/F and other biomarkers.

Quantitative scores (Qanadli and Mastora) and signs of RVD were assessed separately. In the event of disagreement, a consensus reading was performed. The interpretation time for each radiologist was recorded, and the mean value was reported as the manual evaluation time. For statistical analyses, the final values were obtained by consensus.

### 2.4. Subgroup Analysis

To assess whether laboratory parameters and CT signs of right-ventricular strain differed across thrombotic burden strata, patients were dichotomized by the Qanadli obstruction index using a pre-specified threshold of 40% [4,11]: Low Thrombotic Burden (Qanadli < 40%) and High Thrombotic Burden (Qanadli ≥ 40%).

Conversely, to evaluate whether laboratory parameters and thrombotic burden differed across RV strain strata, patients were also divided by the RV/LV short-axis diameter ratio, with a threshold of 1.0: No RV Dysfunction (RV/LV < 1.0) and RV Dysfunction (RV/LV ≥ 1.0). Group comparisons included laboratory variables (e.g., D-dimer, fibrinogen, D-dimer/fibrinogen ratio) and imaging metrics (Qanadli index, RV/LV ratio).

### 2.5. Statistical Analysis

Statistical analysis was performed using the MedCalc program (MedCalc version 11.4.4.0, MedCalc Software bvba, Mariakerke, Belgium).

Descriptive statistics were summarized as mean (standard deviation) for approximately normally distributed variables, median (interquartile range) for non-normally distributed variables, and counts with percentages for categorical data. For the principal variables, 95% confidence intervals (CIs) were reported.

Normality of continuous variables was assessed with the Shapiro–Wilk test.

Spearman’s correlation coefficient (ρ) was used to evaluate the associations between laboratory data (D-d/F, D-dimer and fibrinogen) and imaging metrics (thrombus burden scores, Qanadli% and Mastora%; signs of right ventricular dysfunction, RV/LV ratio, AP/Ao and IVC reflux).

In addition, the correlations between thrombus-burden scores (Qanadli%, Mastora%) and RV strain markers (RV/LV, PA/Ao, IVC reflux) were also examined using Spearman’s ρ.

Steiger’s Z test was applied to compare the strength of correlations between dependent variables with a common outcome.

To examine whether interventricular septal bowing (present or absent on CTPA) was associated with differences in laboratory measures (D-d/F, D-dimer and fibrinogen) and thrombus-burden scores (Qanadli%, Mastora%), we compared groups using the Mann–Whitney U test.

Subgroup comparisons for the predefined strata—Low vs. High Thrombotic Burden (Qanadli < 40% vs. ≥40%) and No RV Dysfunction vs. RV Dysfunction (RV/LV < 1.0 vs. ≥1.0)—were performed using the Mann–Whitney U test for continuous variables (D-dimer, fibrinogen, D-d/F; Qanadli%, Mastora%, RV/LV, PA/Ao) and χ^2^ or Fisher’s exact test for categorical variables, as appropriate. Results for continuous variables are reported as median (IQR), with two-sided *p*-values and a significance threshold of *p* = 0.05.

To evaluate the discrimination of the subgroup definitions, ROC curves were generated for D-dimer and the D-dimer/fibrinogen ratio (D-d/F) against the endpoints Qanadli ≥ 40% and RV/LV ≥ 1.0. Areas under the curve (AUCs) were estimated with 95% confidence intervals using the DeLong method; paired DeLong tests were used to compare AUCs for D-dimer vs. D-d/F on the same endpoint. Optimal cut-offs were identified by maximizing Youden’s J (sensitivity + specificity − 1), and corresponding sensitivity and specificity (with 95% CIs by exact binomial) were reported.

Two logistic regression models were built to identify predictors of high thrombus burden (Qanadli ≥ 40%). Model 1 included age, sex, and the D-dimer/fibrinogen ratio (DFR). Model 2 additionally included the CT-derived RV/LV short-axis diameter ratio (VDx/VSx). Continuous predictors were checked for linearity in the logit model and standardized when necessary; multicollinearity was assessed using variance inflation factors.

## 3. Results

The median D-dimer was 3.1125 µg/mL (2.649 to 3.661) and the median fibrinogen level was 388 mg/dL (351.4524 to 435.1904), resulting in a median D-d/F ratio of 0.7545 (0.6236 to 0.9359) (Table 1).

The median Qanadli score was 37.5% (3 to 42.5, and the median Mastora score was 17.0968% (14.2908 to 22.5806) (Table 1).

Regarding signs of right ventricular dysfunction, the median RV/LV was 0.9 (0.8800 to 0.9385), signs of septal bowing was observed in 33/112 cases (29.5%), contrast reflux into the inferior vena cava (IVC) grade 1 (37.5%), grade 2 (38.4%), grade 3 (24.1%), the median value of AP/Ao was 0.85 (0.83 to 0.8885).

### 3.1. Spearman Analysis

D-d/F showed a statistically significant weak positive correlation with both Qanadli (Spearman ρ = 0.233, 95% CI, 0.0498–0.401; *p* = 0.013) and Mastora scores (Spearman ρ = 0.274, 95% CI, 0.0937–0.438; *p* = 0.0034). D-dimer also showed a weak significant correlation with Qanadli (Spearman ρ = 0.240, 95% CI, 0.0575–0.408; *p* = 0.010) and Mastora scores (Spearman ρ = 0.248, 95% CI, 0.0659–0.415; *p* = 0.008) (Figure 5). However, Steiger’s Z-test revealed no statistically significant difference between the strength of correlations of D-d/F and D-dimer with either Qanadli (Z = 0.18, *p* = 0.857) or Mastora (Z = −0.65, *p* = 0.514).

Regarding correlation analysis between D-dimer, D-d/F and signs of right ventricular dysfunction, Spearman’s rank correlation coefficient between the RV/LV ratio and D-dimer was ρ = 0.312 (95% CI, 0.135–0.471; *p* = 0.0008). Similarly, a moderate statistically significant correlation was found between D-d/F and RV/LV (ρ = 0.335, 95% CI, 0.159–0.490; *p* = 0.0003) (Figure 5). When compared using Steiger’s Z-test, no statistically significant difference was observed between the correlation coefficients of D-dimer and D-d/F with the RV/LV ratio (Z = −0.19, *p* = 0.850).

Conversely, the correlation with fibrinogen was negative and not statistically significant (ρ = −0.136, 95% CI, −0.314–0.0505; *p* = 0.1517). No statistically significant associations were found between AP/Ao and D-dimer or D-d/F.

Both thrombotic burden scores demonstrated stronger associations with the RV/LV ratio. The Qanadli score was strongly correlated with RV/LV (ρ = 0.571, 95% CI, 0.431–0.684; *p* < 0.0001), as was the Mastora score (ρ = 0.620, 95% CI, 0.491–0.723; *p* < 0.0001) (Figure 6).

In addition to the RV diameter ratio analysis, correlation with the severity of inferior vena cava (IVC) reflux on CTPA (graded using the Aviram scale, 1–3) was also performed. Among the laboratory markers, the D-d/F ratio exhibited a significant positive weak correlation with IVC reflux (ρ = 0.247, 95% CI, 0.0649–0.414; *p* = 0.0085), while D-dimer showed a weaker, borderline significant correlation (ρ = 0.182, 95% CI, −0.00347–0.356; *p* = 0.0545). Fibrinogen alone was inversely correlated with IVC reflux severity (ρ = −0.224, 95% CI, −0.394–−0.0406; *p* = 0.0173).

As with the ventricular diameter ratio, CT-based thrombus burden scores demonstrated stronger associations. The Qanadli score demonstrated a strong and statistically significant correlation with IVC reflux (ρ = 0.508, 95% CI, 0.356–0.634; *p* < 0.0001), while the Mastora score showed an even stronger correlation (ρ = 0.570, 95% CI, 0.430–0.683; *p* < 0.0001).

### 3.2. Mann–Whitney Analysis

Mann–Whitney analysis based on the presence or absence of interventricular septal bowing revealed that patients with septal deviation had significantly higher D-dimer levels (median 4.653 vs. 2.738 ng/mL, *p* = 0.0037), as well as elevated D-d/F values (median 1.04 vs. 0.61, *p* = 0.0018). In contrast, no significant differences were observed for fibrinogen (*p* = 0.4213) between the two groups.

Furthermore, patients with septal bowing exhibited significantly higher thrombotic scores. The median Mastora score was 40.6% in patients with septal deviation versus 10.3% in those without (*p* < 0.0001), while the Qanadli score was similarly elevated (median 50% vs. 25%, *p* < 0.0001).

### 3.3. Subgroup Analysis for Qanadli Score

Patients were divided into two groups based on the Qanadli obstruction index threshold of 40% (Low Thrombotic Burden group and High Thrombotic Burden group) [4,11]. Among the 112 patients, 61 (54.5%) were classified as having low thrombotic burden (LTB) (Qanadli < 40%), and 51 (45.5%) as having high thrombotic burden (HTB) (Qanadli ≥ 40%). The median age did not differ significantly between groups (66 vs. 68 years, *p* = 0.2209).

D-dimer levels were significantly higher in the HTB group compared to those of the LTB group (median 3.418 vs. 2.789 ng/mL; *p* = 0.0316). Conversely, fibrinogen levels did not differ significantly between the groups (median 364 vs. 417 mg/dL; *p* = 0.5729). Accordingly, the D-d/F showed an increasing trend in patients with HTB (median 0.63 vs. 0.88, *p* = 0.028) (Figure 7).

No significant difference was observed in INR values between the two groups (median 1.21 vs. 1.30; *p* = 0.1273).

Regarding right ventricular dysfunction, the RV/LV ratio was significantly elevated in the HTB group (median 1.07 vs. 0.87; *p* < 0.0001).

However, no significant difference was observed in the PA/Ao ratio between the two groups (*p* = 0.93).

Receiver operating characteristic (ROC) curve analysis showed that both D-d/F ratio and D-dimer had comparable performance in identifying patients with Qanadli ≥ 40%, with modest areas under the curve (AUC) of 0.621 (95% CI: 0.524–0.711; *p* = 0.0233) and 0.618 (95% CI: 0.522–0.708; *p* = 0.0265), respectively. The difference in AUC between the two markers, as determined by the DeLong test, was not statistically significant (AUC difference: 0.00225; *p* = 0.9256) (Figure 8). The optimal cut-off point for the D-d/F ratio was >0.795, yielding a sensitivity of 58.8% and a specificity of 62.3%, whereas D-dimer > 1.894 µg/mL yielded a sensitivity of 84.3% and a specificity of 37.7%.

### 3.4. Subgroup Analysis for RV/LV Short-Axis Diameter Ratio

Patients were also divided into two groups based on the RV/LV short-axis diameter ratio (VDx/VSx) threshold of 1.0 (No RV Dysfunction group and RV Dysfunction group). Among the 112 patients, 75 were classified as having no RV dysfunction (RV/LV < 1.0), and 37 as having RV dysfunction (RV/LV ≥ 1.0). The median age did not differ significantly between groups (66 vs. 68 years; *p* = 0.2209).

D-dimer concentrations were higher in the RV-dysfunction cohort than in patients without RV dysfunction (median 4.653 vs. 2.644 µg/mL; *p* = 0.0016) (Figure 9). Fibrinogen did not differ meaningfully between groups (median 354 vs. 417 mg/dL; *p* = 0.1621). Consistently, the D-dimer-to-fibrinogen ratio (D-d/F) was increased in the RV-dysfunction group (median 1.04 vs. 0.59; *p* = 0.0009) (Figure 9). INR values showed no significant between-group difference.

Regarding CT thrombus-burden metrics, both Qanadli and Mastora scores were higher in patients with RV dysfunction (Qanadli: median 50 vs. 25, *p* < 0.0001; Mastora: median 40.0 vs. 10.3, *p* < 0.0001) (Figure 9).

ROC analysis demonstrated comparable discrimination of RV/LV ≥ 1.0 for D-d/F and D-dimer, with AUCs of 0.693 (95% CI, 0.599–0.777; *p* = 0.0002) and 0.684 (95% CI, 0.589–0.769; *p* = 0.0006), respectively. The AUC difference by the DeLong test was not significant (ΔAUC = 0.00901; *p* = 0.7201) (Figure 10). The optimal D-d/F threshold (>0.607) yielded 83.8% sensitivity and 52.0% specificity, while D-dimer > 2.849 µg/mL provided 75.7% sensitivity and 54.7% specificity.

### 3.5. Logistic Regression Analysis

Regarding logistic regression analysis, in the initial model including age, sex, and the D-dimer/fibrinogen ratio (DFR), only DFR showed an independent association with high thrombus burden (β = 0.359; OR 1.43, 95% CI 1.01–2.04; *p* = 0.0426), whereas age (OR 1.02; *p* = 0.24) and sex (OR 1.10; *p* = 0.82) were not significant. The model was well calibrated (Hosmer–Lemeshow *p* = 0.88) but displayed modest discrimination (AUC 0.64, 95% CI 0.55–0.73); at a 0.5 probability threshold, overall accuracy was 60.7% with 82.0% specificity and 35.3% sensitivity (Table 2).

After adding the CT-derived RV/LV ratio, model fit improved markedly (LR χ^2^ = 36.98, df = 4; *p* < 0.0001). RV/LV emerged as a strong, independent predictor of Qanadli ≥ 40% (β = 6.95; *p* < 0.0001), translating to an OR of 2.00 per 0.1-unit increase (95% CI, 1.45–2.78). In this extended model, DFR was no longer independently associated with the outcome (OR 1.16, 95% CI 0.88–1.53; *p* = 0.287), and age and sex remained non-significant. Calibration remained acceptable (Hosmer–Lemeshow *p* = 0.087), while discrimination improved to good (AUC 0.796, 95% CI 0.709–0.866). Using a 0.5 threshold, accuracy increased to 74.1%, with specificity 85.3% and sensitivity 60.8% (Table 3).

## 4. Discussion

To the best of our knowledge, this is the first study to assess—within the same cohort of PE patients—the association of the D-dimer/fibrinogen ratio with CT-quantified thrombus burden (Qanadli, Mastora) and CT-derived RV strain (RV/LV).

In this single-centre cohort of patients with acute pulmonary embolism (PE), the D-dimer–to–fibrinogen ratio (D-d/F) showed weak but significant correlations with CT-based thrombotic burden (Qanadli% and Mastora%) and with CT signs of right ventricular dysfunction (RVD), particularly the RV/LV diameter ratio.

Although statistically significant, these modest correlations indicate limited explanatory power at the individual level and should be viewed as descriptive, hypothesis-generating evidence.

The strength of these associations was comparable to that of D-dimer alone and, accordingly, the discriminatory performance of D-d/F and D-dimer for identifying patients with Qanadli ≥ 40% was similar. Clinically, these findings suggest that D-d/F can condense coagulation–fibrinolysis information relevant to risk stratification, yet does not clearly outperform D-dimer for selecting patients with higher clot burden.

These results align with a consolidated literature linking quantitative embolic burden and imaging markers of right-sided strain to early adverse outcomes.

Studies using obstruction scores (Qanadli, Mastora) and 3D volumetric approaches have shown robust associations between thrombus burden, RVD, and prognosis.

In particular, volumetric clot quantification correlates with RV/LV ratio and impending shock, supporting the role of objective measures of embolic load for risk stratification [4,5,12,13]; as well as central/multilobar clot location being associated with RV dysfunction [14].

Imaging parameters of strain also relate to short-term mortality in large ED cohorts, underscoring the primacy of CTPA for early risk assessment [2].

The prospective cohort by van der Meer et al. showed that CTPA markers of right ventricular dysfunction (RV/LV ratio) and the obstruction index assessed on baseline spiral CT were both significant predictors of 3-month mortality, with a markedly higher risk in patients with PAOI ≥ 40% [15]. However, it has been reported that inferior vena cava contrast reflux on CTPA is an independent predictor of 30-day mortality (OR 3.29; *p* = 0.001), whereas obstruction scores and morphometric RVD indices showed inferior performance in risk stratification of acute pulmonary embolism [16].

On the biomarker side, our data confirm prior observations that D-dimer levels track radiologic severity and right-sided strain. Specifically, higher D-dimer levels are associated with greater obstruction (e.g., PAOI/Mastora) and larger RV/LV ratios, and tend to decline in parallel with thrombus resolution on follow-up [17,18,19].

In a multicenter cohort of 674 patients with pulmonary embolism, D-dimer levels > 3000 ng/mL (FEU) were independently associated with 15-day and 3-month mortality, occurring more frequently in the presence of central emboli and comorbidities (active cancer, age > 65 years). The combination of elevated D-dimer and central location approximately doubled the risk of early death (OR = 2.2) [20].

The physiological rationale for considering D-d/F—rather than D-dimer alone—is plausible: D-dimer reflects fibrinolysis of cross-linked fibrin, whereas fibrinogen, an acute-phase reactant and coagulation substrate, can rise with systemic inflammation yet decrease with consumption during active thrombosis. Their ratio may therefore partially normalize inflammatory “noise” on D-dimer and express a relative coagulation–fibrinolysis state.

Consistent with this pathophysiological framework, Kucher et al. demonstrated in 120 patients that factor XIII subunit A levels were significantly lower in confirmed PE cases compared to excluded suspects, decreased progressively with increasing pulmonary obstruction index, and correlated inversely with D-dimer levels (and directly with fibrinogen). These findings suggest coagulative consumption proportional to thrombotic burden and provide mechanistic support for interpreting the D-d/F ratio as an integrated marker of coagulation–fibrinolysis balance [21].

In the emergency setting, a prospective study reported similar AUCs (0.87) for D-d/F and D-dimer to diagnose PE, with slightly higher specificity for D-d/F at sensitivity-maximizing cut-offs; fibrinogen alone was uninformative [8].

In post-partum patients—where D-dimer specificity is notoriously limited—D-d/F has shown diagnostic value and an independent association with PE, suggesting potential to reduce unnecessary CTPA in this subgroup [9]. Beyond VTE, D-d/F has been explored as a prognostic or predictive marker in high-inflammatory or perioperative contexts (e.g., after living-donor liver transplantation), further supporting its role as a composite hemostatic signal [10].

In a hypertensive population with suspected ischemic stroke, Abebe et al. reported comparable AUC values for D-dimer (0.776) and the D-dimer/fibrinogen ratio (0.763), with no significant difference, supporting an additive rather than a substitutive role of the D-d/F ratio [22].

In a 99-patient cohort with acute PE, D-dimer levels correlated independently with CT-based severity, including RV/LV (r = 0.54) and PAOI (r = 0.52). Higher values were observed with more proximal emboli, exhibiting a dose–response pattern for elevated troponin-T and the use of thrombolysis, supporting D-dimer as a proxy for thrombus burden and hemodynamic stress [7].

In our cohort, fibrinogen alone showed no meaningful association with severity indices, whereas D-d/F, despite only weak correlations, retained informativeness for RV/LV, obstruction scores, and (to a lesser extent) IVC reflux.

However, observed discrimination was modest (AUC 0.62–0.69); accordingly, D-d/F should be regarded only as a complementary severity signal, not a stand-alone diagnostic test or a substitute for CTPA or D-dimer.

Stratification by Qanadli ≥ 40% confirmed an expected clinic-radiologic gradient: the high-burden group exhibited higher RV/LV, higher D-dimer, and D-d/F. These observations agree with prior reports linking CT burden to RVD and outcomes. However, the application of specific cut-offs (such as 40%) varies across studies and should be interpreted in the context of predefined endpoints [4].

Our multivariable analyses highlight two complementary messages. First, in a basic model with age and sex, the D-d/F was independently associated with high thrombus burden, suggesting that a hypercoagulable profile captured by D-d/F tracks with clot load. Secondly, once the CT-derived RV/LV short-axis ratio was introduced, RV/LV became the dominant predictor, while D-d/F lost statistical independence. This pattern supports a pathophysiologic sequence in which thrombotic burden translates into hemodynamic strain that is directly measurable on CT; RV/LV, therefore, concentrates much of the information relevant to clot burden.

Finally, beyond RV/LV, our analyses of septal bowing and IVC reflux showed tighter correlations with obstruction scores than with plasma biomarkers, which is coherent with the hemodynamic nature of these CT signs and their closer link to pulmonary circulatory physiology than to clot size alone [2,12].

Several limitations warrant consideration. The retrospective, single-center design and the moderate sample size may have limited our ability to detect associations beyond weak–to–moderate. Selection criteria—most notably the exclusion of patients on prior anticoagulation or with advanced malignancy/end-stage disease—enhance internal validity but may restrict generalizability to the broader PE population. The study focused on imaging severity rather than hard clinical endpoints, so we could not test whether D-d/F adds prognostic information beyond established markers (e.g., troponin/BNP) or beyond D-dimer in multivariable models.

We were unable to provide a conventional baseline clinical chart or adjust for comorbidities, cardiac biomarkers, or PESI/sPESI, as the registry does not capture these variables. Biomarkers were measured only at baseline; we did not evaluate longitudinal trajectories of D-d/F.

Furthermore, we did not perform a formal incremental predictive analysis to determine whether D-d/F improves risk prediction beyond imaging-based models.

## 5. Conclusions

In acute PE, D-d/F correlates modestly yet significantly with CT-derived thrombotic burden and RVD, with performance comparable to D-dimer for identifying patients with more extensive obstruction.

Given its modest performance, D-d/F plays a complementary, not substitutive, role alongside CTPA severity indices and D-dimer, but further prospective studies are needed to establish its clinical utility.

## Figures and Tables

**Figure 1 diagnostics-15-02875-f001:**
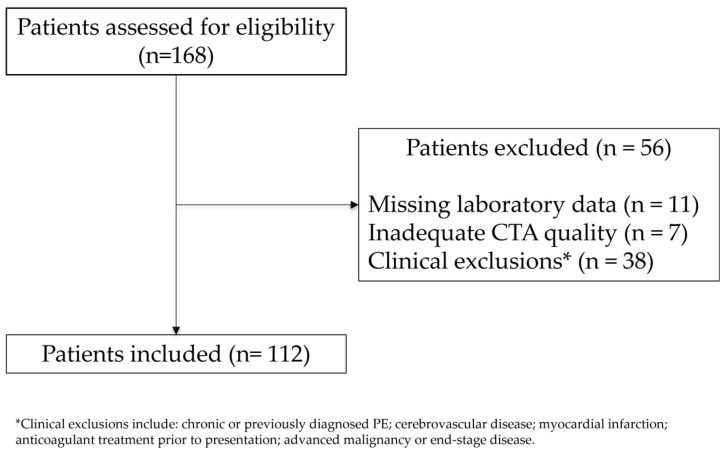
Study flow diagram. Patients assessed for eligibility (January 2022–October 2024), exclusions by primary reason, and final analytic cohort (*n* = 112). Patients with multiple reasons were counted once according to a prespecified hierarchy.

**Figure 2 diagnostics-15-02875-f002:**
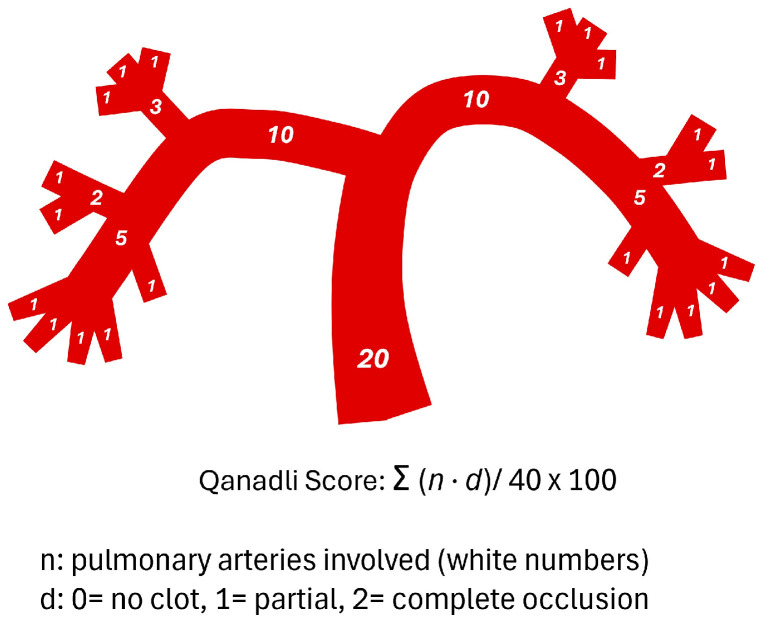
Schematic representation of the pulmonary arterial tree and the Qanadli score.

**Figure 3 diagnostics-15-02875-f003:**
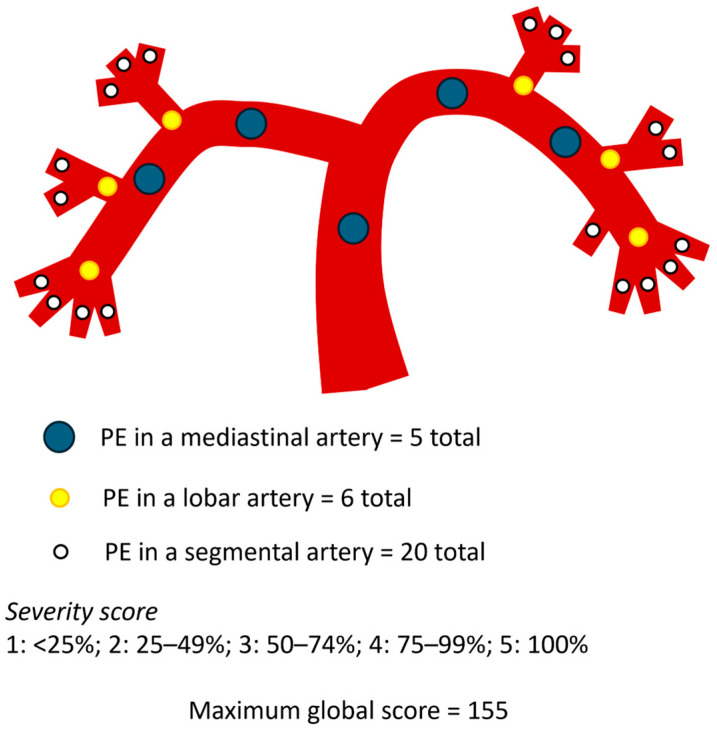
Schematic representation of the pulmonary arterial tree and the Mastora score.

**Figure 4 diagnostics-15-02875-f004:**
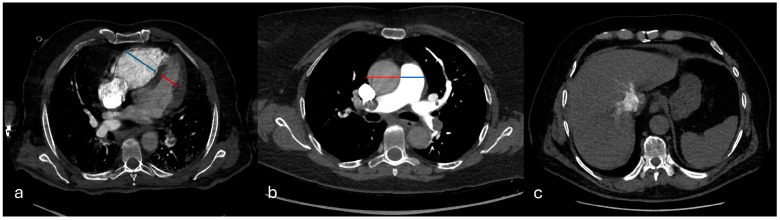
Signs of right ventricular dysfunction. In (**a**), the right-to-left ventricular diameter ratio measured on the same axial slice at the valvular plane, using inner-edge–to-inner-edge calipers; in (**b**), the main pulmonary artery diameter (PA) and ascending aorta (Ao) diameter (PA/Ao ratio); in (**c**), contrast reflux into the inferior vena cava (IVC).

**Figure 5 diagnostics-15-02875-f005:**
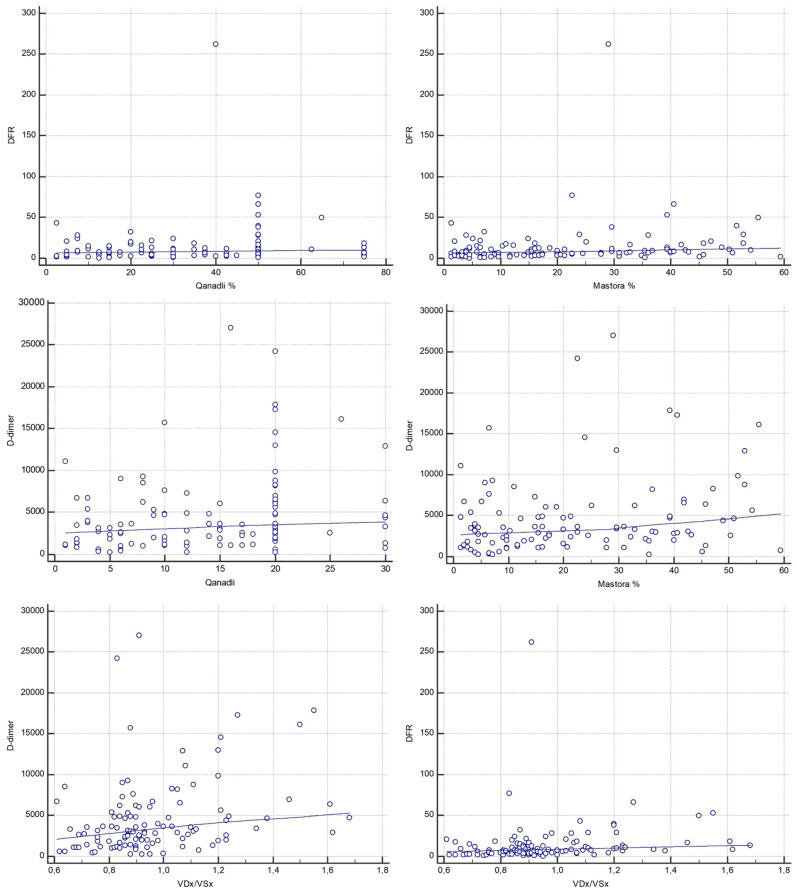
Correlations between laboratory markers (D-d/F and D-dimer), CT obstruction indices (Qanadli and Mastora) and CT-derived RV/LV short-axis ratio (VDx/VSx).

**Figure 6 diagnostics-15-02875-f006:**
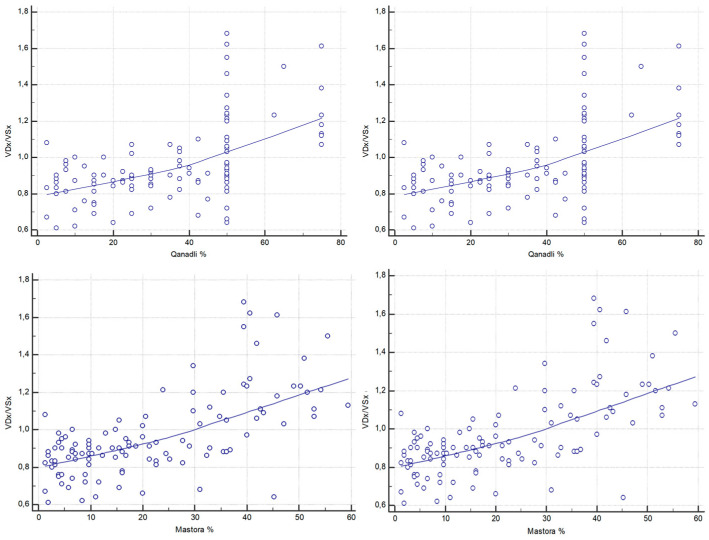
Correlations between CT-derived RV/LV short-axis ratio (VDx/VSx) and CT obstruction indices (Qanadli and Mastora).

**Figure 7 diagnostics-15-02875-f007:**
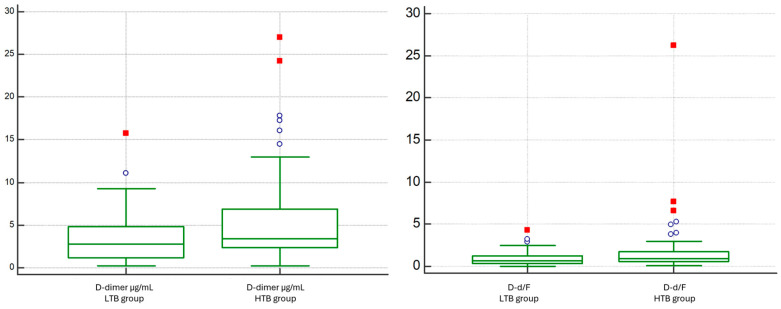
Comparison of D-dimer and D-dimer/Fibrinogen Ratio between Low vs. High Thrombotic Burden (Qanadli < 40% vs. ≥40%).

**Figure 8 diagnostics-15-02875-f008:**
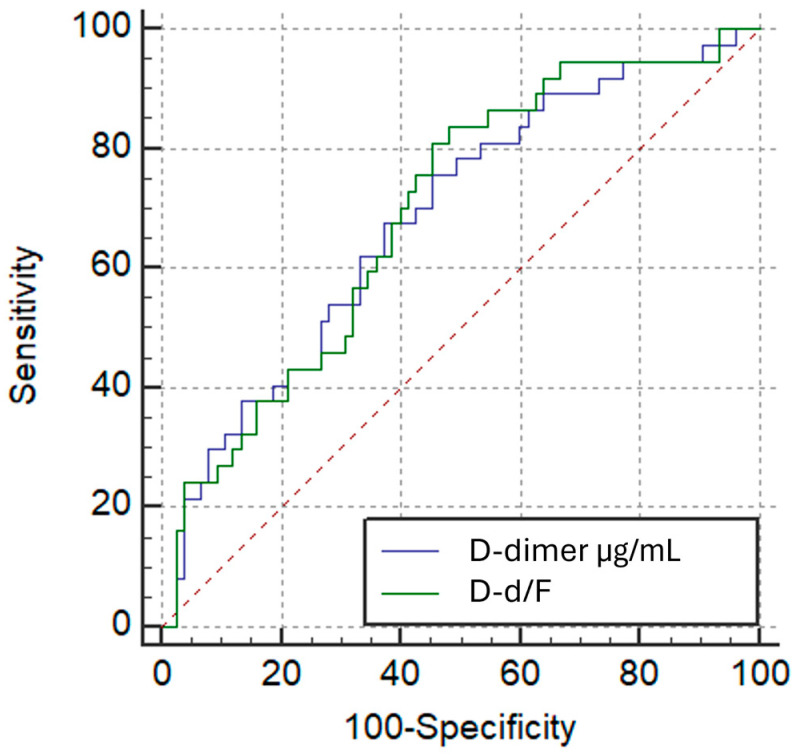
ROC Curves: D-dimer/Fibrinogen Ratio vs. D-dimer for High Thrombotic Burden (Qanadli ≥ 40%).

**Figure 9 diagnostics-15-02875-f009:**
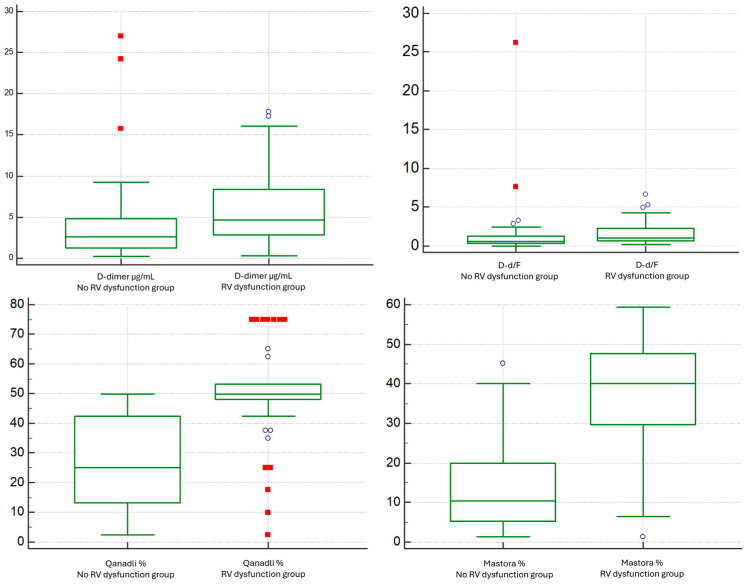
Comparison of D-dimer and D-dimer/Fibrinogen Ratio between the No RV Dysfunction group and the RV Dysfunction group based on the RV/LV short-axis diameter ratio (VDx/VSx) threshold of 1.0.

**Figure 10 diagnostics-15-02875-f010:**
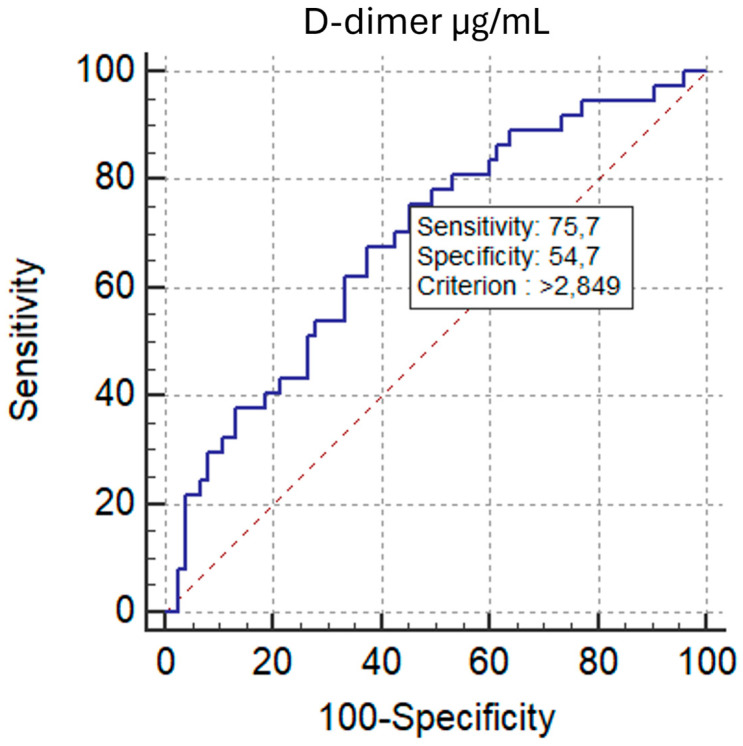
ROC Curves: D-dimer/Fibrinogen Ratio vs. D-dimer for RV Dysfunction group based on the RV/LV short-axis diameter ratio (VDx/VSx) threshold of 1.0.

**Table 1 diagnostics-15-02875-t001:** Characteristics of population study.

	Mean (SD)	Median (95% CI)
Age	65.3839 ± 15.5987	67 (63 to 71)
Sex (male)	60/112	
D-dimer	4.6045 ± 4.7129	3.1125 (2.6495 to 3.6611)
Fibrinogen	434.3036 ± 194.7259	388 (351.4524 to 435.1904)
D-d/F	1.405 ± 2.704	0.754 (0.623 to 0.935)
INR	1.2626 ± 0.2233	1.2550 (1.2000 to 1.3000)
Qanadli Score %	34.6205 ± 19.6541	37.5 (30 to 42.5000)
Mastora Score %	21.7224 ± 16.1504	17.0968 (14.2908 to 22.5806)
RV/LV	0.9615 ± 0.2182	0.9000 (0.8800 to 0.9385)
AP/Ao	0.8399 ± 0.1409	0.8500 (0.8300 to 0.8885)

**Table 2 diagnostics-15-02875-t002:** Logistic regression (Model 1) predicting High Thrombotic Burden (Qanadli ≥ 40%).

Variable	Scale	OR	95% CI
Age	per 1 year	1.0156	0.9897–1.0421
Sex (female vs. male)	—	1.0954	0.5021–2.3900
D-dimer/fibrinogen ratio (D-d/F)	per 1 unit	1.4312	1.0120–2.0242

Model fit: −2LL (null) = 154.371; −2LL (full) = 145.499; χ^2^(3) = 8.872, *p* = 0.0310. Calibration: Hosmer–Lemeshow χ^2^(8) = 3.772, *p* = 0.8771. Discrimination: AUC = 0.644 (SE 0.0519; 95% CI 0.548–0.732). Classification (cut-off *p* = 0.50): overall accuracy 60.71% (Y = 0: 81.97%; Y = 1: 35.29%).

**Table 3 diagnostics-15-02875-t003:** Logistic regression (Model 2) predicting High Thrombotic Burden (Qanadli ≥ 40%).

Variable	Scale	OR	95% CI
Age	per 1 year	0.9989	0.9685–1.0302
Sex (female vs. male)	—	1.1767	0.4819–2.8735
D-dimer/fibrinogen ratio (D-d/F)	per 1 unit	1.1621	0.8812–1.5325
RV/LV ratio	per 1.0 unit	1042.48	40.05–27,134.56

Model fit: −2LL (null) = 154.371; −2LL (full) = 117.396; χ^2^(4) = 36.975, *p* < 0.0001. Calibration: Hosmer–Lemeshow χ^2^(8) = 13.814, *p* = 0.0867. Discrimination: AUC = 0.796 (SE 0.0455; 95% CI 0.709–0.866). Classification (cut-off *p* = 0.50): overall accuracy 74.11% (Y = 0: 85.25%; Y = 1: 60.78%).

## Data Availability

The original contributions presented in this study are included in the article. Further inquiries can be directed to the corresponding author.

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
