# Peer review of "D-Dimer/Fibrinogen Ratio and Radiological Severity Scores in Acute Pulmonary Embolism: Is There Room for a New Thrombus-Burden Marker?"

_diagnostics, 2025, doi:10.3390/diagnostics15222875_

Round 1

Reviewer 1 Report

Comments and Suggestions for Authors

This single-center retrospective study of consecutive adults with CTPA-confirmed acute PE show that in acute PE the D-dimer/fibrinogen ratio (D-d/F), a composite biomarker reflecting the balance between fibrinolysis and coagulation how may therefore partially normalize inflammatory “noise” on D-dimer , tracks CT-derived thrombus burden (clot burden was quantified using Qanadli and Mastora indices) and right-ventricular dysfunction (RVD) to a similar (modest) extent as D-dimer. D-d/F can aid early risk appraisal, CT-based markers—particularly RV/LV—dominate prediction of high clot burden. Of course, the study discussed imaging severity rather than hard clinical endpoints and, because biomarkers were measured at baseline only, this study did not evaluate longitudinal trajectories of D-d/F who could have been of interest.

Author Response

Comment 1: This single-center retrospective study of consecutive adults with CTPA-confirmed acute PE show that in acute PE the D-dimer/fibrinogen ratio (D-d/F), a composite biomarker reflecting the balance between fibrinolysis and coagulation how may therefore partially normalize inflammatory “noise” on D-dimer , tracks CT-derived thrombus burden (clot burden was quantified using Qanadli and Mastora indices) and right-ventricular dysfunction (RVD) to a similar (modest) extent as D-dimer. D-d/F can aid early risk appraisal, CT-based markers—particularly RV/LV—dominate prediction of high clot burden. Of course, the study discussed imaging severity rather than hard clinical endpoints and, because biomarkers were measured at baseline only, this study did not evaluate longitudinal trajectories of D-d/F who could have been of interest.

Response 1: We thank the Reviewer for the accurate summary of our aims and main findings. We agree that CT-based markers—especially RV/LV—remain the dominant predictors of imaging severity, and we explicitly position D-d/F as a complementary, pragmatic biomarker for early appraisal rather than a replacement for imaging. We also agree with the two limitations highlighted. We focused on imaging-based severity (clot burden and RV dysfunction) rather than hard clinical outcomes; and biomarkers were measured only at baseline, so we could not assess longitudinal kinetics of D-d/F.

Reviewer 2 Report

Comments and Suggestions for Authors

Congrats with an excellent rerults obtained. 

112 patients with PE were enrolled into the study, but you mentioned exclusion criteria. Could you add CONSORT 2010 flow diagram.

I did not find the table (or any information) with baseline patient's clinical characterictcs (comorbidities, kidney disfunction,  troponin level, PE severity by PESI ect)

The main question is why did you chose radiological severity score instead of hemodynamic instability to test your hypothesis? 

Author Response

Congrats with an excellent rerults obtained. 

- We thank the Reviewer for the constructive feedback and positive assessment of our results. Below we address each point and describe the corresponding manuscript changes.

112 patients with PE were enrolled into the study, but you mentioned exclusion criteria. Could you add CONSORT 2010 flow diagram.

- Thank you. While CONSORT is intended for randomized trials, we agree that a participant-flow diagram improves transparency in observational studies (as encouraged by STROBE). We have added a CONSORT-style/STROBE-aligned flowchart summarizing screening, eligibility, exclusions, and the final analytic cohort.

I did not find the table (or any information) with baseline patient's clinical characterictcs (comorbidities, kidney disfunction,  troponin level, PE severity by PESI ect)

- We agree that this information is valuable. However, our retrospective imaging and laboratory registry does not systematically record clinical variables at the medical record level (comorbidities, renal function, troponin/BNP, PESI/sPESI). Only imaging (Qanadli, Mastora, RV/LV) and laboratory variables are recorded, which we have reported and described: D-dimer, fibrinogen, and time parameters. Access to complete medical records was not possible, given that our study was retrospective and these variables could not be reconstructed post hoc. We have revised the manuscript to explicitly state this in the Methods section and acknowledge it as a Limitation.

The main question is why did you chose radiological severity score instead of hemodynamic instability to test your hypothesis? 

- We thank the Reviewer for the question. We started with the premise that D-dimer is a diagnostic parameter for pulmonary embolism, and that D-dimer levels are elevated in the presence of acute clots due to simultaneous activation of coagulation and fibrinolysis. Normal D-dimer levels effectively rule out acute pulmonary embolism with a high probability in hemodynamically stable patients. Several CT studies suggest that higher D-dimer levels are associated with a significantly increased clot burden in the pulmonary arteries. Our hypothesis was that a composite biomarker integrating fibrinolysis (D-dimer) and coagulation (fibrinogen) would mirror the anatomical/functional thrombus burden and RV strain at presentation. Radiological indices (Qanadli, Mastora, RV/LV) are objective, reproducible, and directly quantify clot load and strain—the constructs we aimed to reflect with D-d/F. Hemodynamic instability, conversely, is multifactorial (preload, comorbidities, fluids/analgesia, timing of anticoagulation) and was beyond the scope of this imaging-severity study. We now state this rationale explicitly and outline that future prospective work should link D-d/F to shock, vasopressor use, and short-term mortality.

Reviewer 3 Report

Comments and Suggestions for Authors

Thank  you for letting me review the manuscript.

This single-center retrospective study investigates whether the D-dimer/fibrinogen ratio (D-d/F) can serve as a biomarker reflecting CT-quantified thrombus burden and right ventricular dysfunction (RVD) in patients with acute pulmonary embolism (PE).
Using data from 112 patients, the authors compared D-d/F with established imaging indices (Qanadli and Mastora scores, RV/LV ratio).
They found weak but statistically significant correlations between D-d/F and CT-based severity markers, similar in magnitude to those observed for D-dimer.
The study concludes that D-d/F may assist in early risk appraisal, but CT-derived metrics—especially RV/LV ratio—remain superior predictors.

Major Points

#1

The physiological rationale of D-d/F is valid, but I believe that prior studies have already evaluated this ratio in PE. The manuscript should better clarify what new knowledge is added beyond showing correlations comparable to D-dimer.

#2

Reported associations (AUC = 0.62–0.69) indicate weak diagnostic performance. The discussion should more clearly acknowledge this limitation and avoid overinterpreting clinical relevance.

#3

The extremely large odds ratio for RV/LV (OR ≈ 1042) suggests a scaling or input error. Please confirm whether RV/LV ratio was standardized, log-transformed, or incorrectly scaled in the regression model.

#4

The Methods section should include a patient flowchart showing numbers screened, excluded, and analyzed. Key exclusion criteria (malignancy, anticoagulation, missing data) are listed, but their actual counts are missing.

#5 Incremental Value Not Demonstrated

The conclusion implies that D-d/F might complement imaging, but no incremental predictive analysis was performed. To justify its utility, such an analysis—or at least a clear statement acknowledging its absence—is necessary.

 Minor Points

#1

Use periods instead of commas for decimals (e.g., “3.11 µg/mL” instead of “3,11”).

The abbreviation “VDx/VSx” should be unified as “RV/LV” throughout for clarity.

#2

Reference #7 and #23 duplicate the same citation (Ghanima et al., 2007) — please remove one. Simplify long sentences in the Discussion to enhance readability; MDPI prefers concise phrasing.

Comments on the Quality of English Language

The grammar and collocations in the manuscript should be reviewed by a native English speaker.

Author Response

Thank  you for letting me review the manuscript.

This single-center retrospective study investigates whether the D-dimer/fibrinogen ratio (D-d/F) can serve as a biomarker reflecting CT-quantified thrombus burden and right ventricular dysfunction (RVD) in patients with acute pulmonary embolism (PE).
Using data from 112 patients, the authors compared D-d/F with established imaging indices (Qanadli and Mastora scores, RV/LV ratio).
They found weak but statistically significant correlations between D-d/F and CT-based severity markers, similar in magnitude to those observed for D-dimer.
The study concludes that D-d/F may assist in early risk appraisal, but CT-derived metrics—especially RV/LV ratio—remain superior predictors.

Major Points

#1

The physiological rationale of D-d/F is valid, but I believe that prior studies have already evaluated this ratio in PE. The manuscript should better clarify what new knowledge is added beyond showing correlations comparable to D-dimer.

  • We thank the Reviewer for this observation. We agree that the D-dimer/fibrinogen ratio (D-d/F) has been examined in prior PE studies. The novel contribution of our work is not to “rediscover” D-d/F, but to test its association with objective CTPA severity markers within the same cohort, through a head-to-head comparison of D-d/F and D-dimer against CT obstruction scores (Qanadli%, Mastora%) and CT-derived right-ventricular strain (RV/LV). We additionally performed formal tests of dependent correlations to determine whether D-d/F truly outperforms D-dimer. Prior investigations have mainly assessed D-d/F for overall diagnosis or clinical outcomes (mortality/instability), typically without a standardized benchmark against CT severity scores or formal comparative statistics between D-d/F and D-dimer within the same cohort. To the best of our knowledge, as stated in the introduction, no previous study has specifically evaluated the relationship between D-d/F and CT-quantified thrombus burden (e.g., Qanadli%, Mastora%) or CT-derived RV strain (RV/LV) within a single cohort.

#2

Reported associations (AUC = 0.62–0.69) indicate weak diagnostic performance. The discussion should more clearly acknowledge this limitation and avoid overinterpreting clinical relevance.

  • We thank the reviewer for their comments. We agree. As already stated in the Introduction, Results, and Discussion, we characterize the associations and discrimination as weak/moderate and do not propose any surrogate diagnostic role for D-d/F. To address this observation even more explicitly, we have further emphasized that the AUC value is modest, both in the results and in the discussion and conclusion, to avoid any overinterpretation.

#3

The extremely large odds ratio for RV/LV (OR ≈ 1042) suggests a scaling or input error. Please confirm whether RV/LV ratio was standardized, log-transformed, or incorrectly scaled in the regression model.

  • Thank you for pointing this out. We re-checked the dataset, coding, and model pipeline and found no input or data-merging errors. The very large OR (≈ 1042) arose from the unit of measurement used in the logistic model—i.e., the effect was reported per 1.0-unit increase in RV/LV. Given that RV/LV values in our cohort typically range around ≈0.7–1.6, a one-unit increase represents a very large clinical shift, which naturally inflates the OR. To provide clinically interpretable effect sizes, in the Results section we already reported the association per +0.1-unit increase of RV/LV: OR = 2.00 with 95% CI 1.45–2.78. This value is the direct rescaling of the per–1.0 unit coefficient. We have corrected the typographical error in the text where “1.45” was inadvertently written as “145”.

#4

The Methods section should include a patient flowchart showing numbers screened, excluded, and analyzed. Key exclusion criteria (malignancy, anticoagulation, missing data) are listed, but their actual counts are missing.

  • Thank you. We agree that a participant-flow diagram improves transparency in observational studies (as encouraged by STROBE). We have added a CONSORT-style/STROBE-aligned flowchart summarizing screening, eligibility, exclusions, and the final analytic cohort.

#5 Incremental Value Not Demonstrated

The conclusion implies that D-d/F might complement imaging, but no incremental predictive analysis was performed. To justify its utility, such an analysis—or at least a clear statement acknowledging its absence—is necessary.

  • We appreciate the reviewer’s observation. We wish to clarify that evaluating risk association or incremental predictive value was not among the aims of this study. Accordingly, we did not perform incremental value analyses. To prevent any misunderstanding, we have revised the text to avoid implying incremental utility and to explicitly acknowledge this limitation.

 Minor Points

#1

Use periods instead of commas for decimals (e.g., “3.11 µg/mL” instead of “3,11”).

The abbreviation “VDx/VSx” should be unified as “RV/LV” throughout for clarity.

  • We thank the reviewer, we have made the requested changes.

#2

Reference #7 and #23 duplicate the same citation (Ghanima et al., 2007) — please remove one. Simplify long sentences in the Discussion to enhance readability; MDPI prefers concise phrasing.

  • We thank the reviewer, the duplicate citation has been deleted.

Reviewer 4 Report

Comments and Suggestions for Authors

Dear authors,

This manuscript presents a well-structured retrospective study exploring the relationship between the D-dimer/fibrinogen ratio and CT-derived thrombus burden and right ventricular dysfunction in patients with acute pulmonary embolism. The topic is clinically relevant, as it bridges imaging-based severity scoring with biochemical markers, potentially contributing to earlier risk stratification in emergency settings. The paper is generally well written and methodologically sound, but certain aspects could benefit from clarification, deeper discussion, and editorial refinement.

The title accurately reflects the study scope and hypothesis.
However:

  • The abstract (lines 17–44) is dense and could benefit from simplified phrasing and clearer emphasis on the main findings and implications.
    For example, line 43: “CT-based markers—particularly RV/LV—dominate prediction of high clot burden” could be expanded to indicate why this is clinically relevant.
  • Please clarify whether the D-d/F ratio is intended as a diagnostic adjunct or a prognostic tool; this distinction is not consistently maintained throughout the paper.

The introduction (lines 52–82) provides a solid rationale for the study and summarizes the current state of evidence regarding D-dimer and fibrinogen.
Suggestions:

  • Add a clear research hypothesis statement at the end of the introduction (e.g., “We hypothesized that the D-d/F ratio would correlate more strongly with CT thrombus burden and RVD than D-dimer alone”).
  • Consider integrating a short paragraph on the biological plausibility of using D-d/F (lines 72–79) with citations emphasizing its previous diagnostic and prognostic roles in thrombotic disease.
  • You may also note that prior literature on D-d/F in acute PE remains limited to small cohorts or specific populations (e.g., postpartum women, ICU settings).

The methodology is detailed and transparent, with proper descriptions of imaging and laboratory protocols. Commendably, the authors described both Qanadli and Mastora indices with illustrative figures (Figures 1–2, page 7) and clarified RVD assessment criteria (page 8).
Nevertheless:

  • Please clarify whether the imaging readers were blinded to the D-d/F data (lines 152–157 suggest blinding, but specify for all markers).
  • The inclusion/exclusion criteria (lines 95–101) are appropriate, but it would be useful to indicate how many patients were excluded for each reason.

The results are presented with good structure, supported by clear figures (Figures 4–9) and tables (Tables 1–3).

While correlations between D-d/F and CT scores (ρ = 0.23–0.27) are statistically significant, their clinical strength is weak. Please discuss this in the text (perhaps lines 194–200 and 353–360) to temper overinterpretation.

  • Add confidence intervals for correlations and regression coefficients where possible.
  • While correlations between D-d/F and CT scores (ρ = 0.23–0.27) are statistically significant, their clinical strength is weak. Please discuss this in the text (perhaps lines 194–200 and 353–360) to temper overinterpretation.

The conclusions (lines 463–471) are balanced, but could be more concise and impact-oriented. For instance:

“D-d/F correlates modestly but significantly with CT-derived thrombus burden and right ventricular strain. It may complement imaging-based assessment, but further prospective studies are needed to establish its clinical utility.”

Author Response

Dear authors,

This manuscript presents a well-structured retrospective study exploring the relationship between the D-dimer/fibrinogen ratio and CT-derived thrombus burden and right ventricular dysfunction in patients with acute pulmonary embolism. The topic is clinically relevant, as it bridges imaging-based severity scoring with biochemical markers, potentially contributing to earlier risk stratification in emergency settings. The paper is generally well written and methodologically sound, but certain aspects could benefit from clarification, deeper discussion, and editorial refinement.

  • Thank you for the thorough and constructive review. We have addressed each point below and revised the manuscript accordingly.

The title accurately reflects the study scope and hypothesis.

  • We appreciate the positive feedback.

However:

The abstract (lines 17–44) is dense and could benefit from simplified phrasing and clearer emphasis on the main findings and implications.
For example, line 43: “CT-based markers—particularly RV/LV—dominate prediction of high clot burden” could be expanded to indicate why this is clinically relevant.

  • We simplified the abstract and expanded the clinical relevance. We also avoided language implying prediction/utility beyond our aims.

Please clarify whether the D-d/F ratio is intended as a diagnostic adjunct or a prognostic tool; this distinction is not consistently maintained throughout the paper.

  • We thank the reviewer. Our study was not designed to assess diagnostic accuracy or prognostic performance. Our aim was descriptive: to examine associations between D-d/F and CT-derived measures. For this reason, we have removed the term "diagnostic performance" from the text. D-d/F is examined as an exploratory biomarker in relation to CT-derived thrombus burden and right ventricular strain; the study was not designed to test diagnostic or prognostic performance.

The introduction (lines 52–82) provides a solid rationale for the study and summarizes the current state of evidence regarding D-dimer and fibrinogen.
Suggestions:

Add a clear research hypothesis statement at the end of the introduction (e.g., “We hypothesized that the D-d/F ratio would correlate more strongly with CT thrombus burden and RVD than D-dimer alone”).

  • We thank the reviewer for the comment, we have added the suggested sentence at the end of the introduction.

Consider integrating a short paragraph on the biological plausibility of using D-d/F (lines 72–79) with citations emphasizing its previous diagnostic and prognostic roles in thrombotic disease.

  • We greatly appreciate the reviewer's comment, so we have added a pathophysiological explanation of the role of Dd/F in the introduction.

You may also note that prior literature on D-d/F in acute PE remains limited to small cohorts or specific populations (e.g., postpartum women, ICU settings).

  • We have included the reviewer's suggestion in the introduction, highlighting the limited existence of studies on dd/f and pulmonary embolism and the need for further investigations.

The methodology is detailed and transparent, with proper descriptions of imaging and laboratory protocols. Commendably, the authors described both Qanadli and Mastora indices with illustrative figures (Figures 1–2, page 7) and clarified RVD assessment criteria (page 8).
Nevertheless:

Please clarify whether the imaging readers were blinded to the D-d/F data (lines 152–157 suggest blinding, but specify for all markers).

  • We thank the reviewer for their comment. Although already specified, we have reemphasized in the text that all image readers were blinded to laboratory results, including D-d/F and other biomarkers.

The inclusion/exclusion criteria (lines 95–101) are appropriate, but it would be useful to indicate how many patients were excluded for each reason.

  • We have added a CONSORT-style/STROBE-aligned flowchart summarizing screening, eligibility, exclusions, and the final analytic cohort.

The results are presented with good structure, supported by clear figures (Figures 4–9) and tables (Tables 1–3).

While correlations between D-d/F and CT scores (ρ = 0.23–0.27) are statistically significant, their clinical strength is weak. Please discuss this in the text (perhaps lines 194–200 and 353–360) to temper overinterpretation.

  • Thank you for this suggestion. In the Results section we already characterize these associations as weak and have kept that section strictly descriptive. To address the reviewer’s point, we have retained the descriptive tone in Results and expanded the Discussion (at the suggested locations) to explicitly state the limited clinical value of the observed effect sizes and to caution against overinterpretation.

Add confidence intervals for correlations and regression coefficients where possible.

  • We thank the reviewer for the comment. We have added confidence intervals.

While correlations between D-d/F and CT scores (ρ = 0.23–0.27) are statistically significant, their clinical strength is weak. Please discuss this in the text (perhaps lines 194–200 and 353–360) to temper overinterpretation.

  • Thank you for this suggestion. In the Results section, we already characterize these associations as weak and have kept that section strictly descriptive. To address the reviewer’s point, we have retained the descriptive tone in Results and expanded the Discussion (at the suggested locations) to explicitly state the limited clinical value of the observed effect sizes and to caution against overinterpretation.

The conclusions (lines 463–471) are balanced, but could be more concise and impact-oriented. For instance:

“D-d/F correlates modestly but significantly with CT-derived thrombus burden and right ventricular strain. It may complement imaging-based assessment, but further prospective studies are needed to establish its clinical utility.”

  • We thank the Reviewer for the comment and suggestion. We have revised and streamlined the conclusion accordingly, also aligning it with the feedback provided by the other reviewers.

Reviewer 5 Report

Comments and Suggestions for Authors

Although the topic is potentially interesting, the paper presents several major weaknesses that limit its scientific value in its current form.

First, the rationale and clinical applicability of assessing biomarkers such as the D-dimer/fibrinogen ratio in patients who are, by definition, already undergoing CT pulmonary angiography (CTPA) for diagnostic purposes, is unclear. The authors should clearly explain what added diagnostic or prognostic value these biomarkers provide beyond imaging findings.

Second, the study lacks a well-defined primary outcome and clear methodological structure. It is difficult to interpret the results without a precise statement of what the authors aimed to demonstrate. The presentation of results, though rich in data, is confusing and reflects the methodological ambiguity.

Furthermore, the discussion is excessively long and verbose, lacking focus and synthesis. It should be condensed and aligned with the actual findings, emphasizing what is genuinely new or relevant in clinical practice.

Comments on the Quality of English Language

A thorough language revision is also needed to improve clarity and readability.

Author Response

Although the topic is potentially interesting, the paper presents several major weaknesses that limit its scientific value in its current form.

First, the rationale and clinical applicability of assessing biomarkers such as the D-dimer/fibrinogen ratio in patients who are, by definition, already undergoing CT pulmonary angiography (CTPA) for diagnostic purposes, is unclear. The authors should clearly explain what added diagnostic or prognostic value these biomarkers provide beyond imaging findings.

  • We thank the reviewer for his comments and for allowing us to explain the purpose of our study. We want to present our perspective respectfully. We started from the concept that, in hemodynamically stable patients with suspected PE, international guidelines recommend D-dimer testing as part of the diagnostic algorithm to decide whether to proceed to imaging; CTPA is performed when D-dimer is positive or when pre-test probability is high. Our retrospective cohort, by design, includes only CTPA-confirmed PE, where diagnostic rule-out is no longer the question. Our study addressed a different clinical question: within confirmed PE, does the D-dimer/fibrinogen ratio (D-d/F) track thrombus burden and right-ventricular dysfunction (RVD) on CT—and does it perform comparably to, or better than, D-dimer alone for these severity surrogates? This is clinically relevant because D-dimer has been linked to clot burden and RV strain; we tested whether D-d/F might better reflect this pathophysiology. We do not claim added diagnostic value over CTPA, nor do we claim incremental predictive utility; our analyses are descriptive/severity-oriented by design.

Second, the study lacks a well-defined primary outcome and clear methodological structure. It is difficult to interpret the results without a precise statement of what the authors aimed to demonstrate. The presentation of results, though rich in data, is confusing and reflects the methodological ambiguity.

  • We thank the Reviewer for this comment. We want to emphasize that methodological rigor is central to our practice. We had a priori a clearly defined primary aim and outcome, stated in the Aim/Methods: to evaluate whether D-d/F varies with CT-quantified thrombus burden (Qanadli%, Mastora%) and CT signs of RVD (RV/LV ratio, septal bowing, IVC reflux) in patients with CTPA-confirmed acute PE. The Methods detail the cohort (consecutive, single-centre; January 2022–October 2024), laboratory definitions, CT protocols, statistical plan (Spearman’s ρ with Steiger’s Z for between-marker comparisons; ROC for Qanadli ≥40% and RV/LV ≥1.0; two exploratory logistic models: Model-1 [age, sex, D-d/F] and Model-2 adding RV/LV). The Results then follow this plan step by step. To clarify our hypothesis, we have inserted it at the end of the study objective in the introduction.

Furthermore, the discussion is excessively long and verbose, lacking focus and synthesis. It should be condensed and aligned with the actual findings, emphasizing what is genuinely new or relevant in clinical practice.

  • We agree to streamline. We have condensed the Discussion, removed redundancies, and focused on three points: what is new—that D-d/F shows modest, statistically significant associations with CT clot burden and RVD comparable to D-dimer; what is not claimed—we do not assert diagnostic replacement of imaging nor incremental predictive value; how this fits with current literature on CT-based severity markers (e.g., RV/LV, obstruction scores). We also tempered interpretation, emphasizing limitations and the need for prospective, hypothesis-driven studies.

Round 2

Reviewer 2 Report

Comments and Suggestions for Authors

No

Author Response

Thank you.

Reviewer 3 Report

Comments and Suggestions for Authors

I appreciate the opportunity to review the revised manuscript. The authors have addressed all my previous comments satisfactorily. I have no further concerns, and I believe the manuscript is now suitable for publication.

Author Response

Thank you.

Reviewer 4 Report

Comments and Suggestions for Authors

The authors have thoroughly addressed all of my previous comments and provided satisfactory clarifications to the raised queries. The revised manuscript has been substantially improved in clarity, structure, and scientific rigor. I find that the current version meets the journal’s standards and is now suitable for publication in its present form.

Author Response

Thank you.

Reviewer 5 Report

Comments and Suggestions for Authors

Authors were unable to satisfactory address my comments

Comments on the Quality of English Language

A thorough language revision is also needed to improve clarity and readability.

Author Response

Please find our answers to each of your comments in the attached document.
